# Effect of Light Conditions, *Trichoderma* Fungi and Food Polymers on Growth and Profile of Biologically Active Compounds in *Thymus vulgaris* and *Thymus serpyllum*

**DOI:** 10.3390/ijms25094846

**Published:** 2024-04-29

**Authors:** Kamila Kulbat-Warycha, Justyna Nawrocka, Liliana Kozłowska, Dorota Żyżelewicz

**Affiliations:** 1Institute of Food Technology and Analysis, Faculty of Biotechnology and Food Sciences, Lodz University of Technology, Stefanowskiego 4/10 St., 90-924 Lodz, Poland; 2Department of Plant Physiology and Biochemistry, Faculty of Biology and Environmental Protection, University of Lodz, Banacha 12/16 St., 90-237 Lodz, Poland; justyna.nawrocka@biol.uni.lodz.pl (J.N.); liliana.kozlowska@edu.uni.lodz.pl (L.K.)

**Keywords:** *Thymus vulgaris*, *Thymus serpyllum*, *Trichoderma*, food polymers

## Abstract

The research investigates the influence of different lighting conditions and soil treatments, in particular the application of food polymers separately and in combination with spores of *Trichoderma* consortium, on the growth and development of herbs—*Thymus vulgaris* and *Thymus serpyllum*. The metabolic analysis focuses on detecting changes in the levels of biologically active compounds such as chlorophyll a and b, anthocyanins, carotenoids, phenolic compounds (including flavonoids), terpenoids, and volatile organic compounds with potential health-promoting properties. By investigating these factors, the study aims to provide insights into how environmental conditions affect the growth and chemical composition of selected plants and to shed light on potential strategies for optimising the cultivation of these herbs for the improved quality and production of bioactive compounds. Under the influence of additional lighting, the growth of *T. vulgaris* and *T. serpyllum* seedlings was greatly accelerated, resulting in an increase in shoot biomass and length, and in the case of *T. vulgaris*, an increase in carotenoid and anthocyanin contents. Regarding secondary metabolites, the most pronounced changes were observed in total antioxidant capacity and flavonoid content, which increased significantly under the influence of additional lighting. The simultaneous or separate application of *Trichoderma* and food polymers resulted in an increase in flavonoid content in the leaves of both *Thymus* species. The increase in terpenoid content under supplemental light appears to be related to the presence of *Trichoderma* spores as well as food polymers added to the soil. However, the nature of these changes depends on the thyme species. Volatile compounds were analysed using an electronic nose (E-nose). Eight volatile compounds (VOCs) were tentatively identified in the vapours of *T. vulgaris* and *T. serpyllum*: α-pinene, myrcene, α-terpinene, γ-terpinene; 1,8-cineole (eucalyptol), thymol, carvacrol, and eugenol. Tendencies to increase the percentage of thymol and γ-terpinene under supplemental lighting were observed. The results also demonstrate a positive effect of food polymers and, to a lesser extent, *Trichoderma* fungi on the synthesis of VOCs with health-promoting properties. The effect of *Trichoderma* and food polymers on individual VOCs was positive in some cases for thymol and γ-terpinene.

## 1. Introduction

*Thymus vulgaris* L. is the most commonly used and studied species of the genus *Thymus* of the *Lamiaceae* family, which is widely used for culinary, medicinal, and cosmetic purposes. As a culinary herb, it has an intense, pleasant smell and a characteristic, slightly burning taste. For this reason, it is commonly used in fresh or dried form as a spice [1,2]. Plants of the genus *Thymus* contain compounds such as bioactive phenolic monoterpenes: thymol and its isomer carvacrol, which possess strong antibacterial, antiviral, and antifungal properties. Thyme essential oil and thymol alone have been demonstrated to exhibit strong antibacterial [3] and antifungal activity against a range of species, including *Aspergillus*, *Penicillium*, *Cladosporium*, *Rhizopus*, and *Mucor* [4]. Additionally, thyme essential oil has been shown to possess antiviral properties against the influenza virus, and virucidal activity against both the acyclovir-sensitive and acyclovir-resistant strain of herpes simplex virus type 1 (HSV-1) [5,6]. Thyme is rich in antioxidants. Recent reports suggest that supplementing the diet with thyme essential oil strengthens the immune system, which together with its antiviral effects can reduce the symptoms of COVID-19 [7]. Some studies have indicated that certain compounds in thyme may possess anti-parasitic properties [8]. Research on experimental animals has demonstrated the hepatoprotective effects of thyme extract [9]. Furthermore, some studies have suggested that thyme’s monoterpenes may possess neuroprotective properties, manifested by the inhibition of oxidative stress in the brain [10], anxiolytic effects [11], and anti-depressant effects [12]. Although wild thyme (*Thymus serpyllum* L.) is a species with numerous advantages, it has not been widely used and has not received much attention in scientific research. Wild thyme is a rich source of proteins, vitamins (A, C, E), minerals (K, Ca, Mg, Si, and Fe), and thymol [13,14]. Studies on Wistar rats have demonstrated the potent antioxidant and antihypertensive properties of *T. serpyllum*, which was rich in polyphenolic compounds with the dominant rosmarinic and caffeic acids [15]. A similar antihypertensive effect was observed in rats supplemented with *T. vulgaris* [16].

It is well established that the chemical composition of thyme essential oils and the resulting properties are significantly influenced by a range of factors, including the growth environment and agronomic management practices [17]. Recently, there has been a growing interest in sustainable, eco-friendly organic farming practices that limit the use of synthetic fertilisers, pesticides, and herbicides. Medicinal and aromatic plants represent one of the top 10 crops that are recommended for cultivation under the organic regime [18,19]. Although there is evidence that organic agriculture can result in the production of more valuable plant products, there are still few methods of growing plants using beneficial microorganisms, biofertilisers, or products generated within the circular economy, dedicated to the cultivation of plants of particular medical and industrial importance, including herbs.

Abiotic stresses, such as high light intensity, can significantly impact secondary metabolism in plants and the composition of bioactive compounds. High light intensity can induce oxidative stress, which can trigger various biochemical pathways, leading to changes in secondary metabolite production. Such alterations are frequently adaptive responses designed to enhance plant resilience to environmental stresses and to maintain cellular homeostasis [20]. One way in which high light-intensity stress affects secondary metabolism is through the activation of the phenylpropanoid pathway responsible for the synthesis of phenolic compounds, including flavonoids, anthocyanins, and lignin. These compounds are essential for plant defence against light stress, UV radiation and oxidative stress [21]. In response to excessive light intensity, plants may increase the production of flavonoids and anthocyanins as a protective mechanism against photo oxidative damage [22]. Another pathway that may be influenced by high light-intensity stress is the terpenoid pathway. Terpenoids, including essential oils components, carotenoids, and steroids, play a multitude of roles in plant defence, communication, and the attraction of pollinators. High light intensity can induce the synthesis of certain terpenoids, such as carotenoids, which act as antioxidants and protect chlorophyll from photodamage [20].

In addition to their importance in plant stress physiology, secondary metabolites have significant practical applications in the pharmaceutical and food industries. The production of these compounds is relatively low and is dependent on the physiological state of the plant and its growth conditions. By understanding the relationship between secondary metabolism and the growth conditions of given herbal species, we can plan the conditions to support the growth and development of plants and intensify the content of compounds with the desired potential [20].

The genus *Trichoderma* contributes to a large number of different, multifunctional fungi widely used in agricultural applications due to their biological control activity. Some *Trichoderma* spp. significantly suppress the growth of plant pathogenic microorganisms and regulate the rate of plant growth [23]. *Trichoderma* spp. can be utilised in the decomposition of waste/organic materials and the detoxification of polluted areas. This is due to various organic polymer-degrading enzymes that are released by *Trichoderma* spp., including xylanases, cellulases, glucanases, and glucosidases [24]. Selected *Trichoderma* strains act as natural decomposition agents for organic material in bioremediation [24,25]. Other strains are able to detoxify pesticides and herbicides or are alternatives to expensive chemical fertilisers [26,27,28]. The success of *Trichoderma* spp. can be attributed to their ability to accelerate plant growth, nutrient uptake, and the ability to modify the rhizosphere. Most findings indicate that *Trichoderma* spp. improve overall plant health by creating a favourable environment and producing a large amount of secondary metabolites [23]. The efficiency of products containing various species or strains may vary within different fields and climate conditions. The discovery of more *Trichoderma* spp. that can positively affect crops, vegetables, fruits, and herbs may support their use as new eco-friendly and natural biofertilisers, potentially replacing the use of synthetic/chemical fertilisers in agriculture [23].

Natural polymeric substances, such as cellulose, starch, arabic and guar gum, alginate, chitosan, xanthan, etc., are extensively used in a variety of fields such as the food, textile, paper, cosmetic, and pharmaceutical industries. They are well known for their biocompatibility, non-toxicity, and biodegradability [29,30,31,32]. As a food ingredient, the applications of natural polymeric substances are based on their different properties, including thickening, gelling, emulsifying, stabilising, and film forming [33,34]. Recently, there has been a growing interest in certain polymeric substances in agriculture as viable alternatives to traditional soil conditioning [29]. Usually, the substances converted in the form of hydrogels are suitable for use in agricultural applications due to their susceptibility to degradation by physical, chemical, and microbial agents [32]. In agriculture, the use of polymers has been shown to improve water-holding capacity, thereby promoting the growth and quality of crops. Furthermore, polymers in hydrogels have the potential to improve the efficiency of fertiliser use, reduce the contamination of groundwater, enhance the physical properties and microbial activity of soil, and facilitate the germination and establishment of seedlings [34]. The aforementioned studies have employed the use of specially synthesised hydrogels, applied individually. However, no studies have been conducted in which hydrogel components, such as alginates or gums, were used in combination with microorganisms belonging to biocontrol agents (BCAs), including *Trichoderma*. To our knowledge, no studies have investigated the effect of different light conditions in combination with soil application of BCAs and biofertilisers on the functioning and generation of secondary metabolites, which determine the antioxidant capacity and health-promoting properties of *T. vulgaris* and *T. serpyllum* herbs.

Taking into account all of the above information, this research aims to investigate the effects of different cultivation conditions, including different lighting and soil treatment with food polymers, separately and in combination with *Trichoderma* BCAs on the growth and quality characteristics, as well as the metabolism, of *T. vulgaris* and *T. serpyllum* herbs. We hypothesised that light in combination with soil application of *Trichoderma* and/or food polymers would have a positive effect on the secondary metabolism of thyme plants. To verify this hypothesis, we performed metabolic studies, which included the detection of changes in the content of photosynthetic pigments, chlorophyll a (Chl a), chlorophyll b (Chl b), and carotenoids, as well as the analysis of secondary metabolites: phenolic compounds, including flavonoids and anthocyanins, and terpenoids, as well as volatile organic compounds of health significance. On the basis of the results obtained, we tried to determine which variety of thyme responded more favourably to the factors applied.

## 2. Results

### 2.1. Growth of T. vulgaris and T. serpyllum

Initially, we examined the impact of lighting and soil supplementation with microorganisms classified as BCAs and a mixture of food polymers, including alginate, guar gum, and xanthan gum on plant fresh biomass and the length of shoots. Considering the results of two-way ANOVA, lighting significantly affected both parameters in *T. vulgaris* and *T. serpyllum*. Additionally, the length of the shoots of *T. vulgaris* was affected by the interaction of lighting and soil treatment, whereas soil treatment separately significantly affected both parameters only in *T. serpyllum* (Appendix A). Regarding the light conditions, under a light intensity of 174 and 600 µmol·m^−2^ s^−1^, an increase in the shoot biomass of *T. vulgaris* and *T. serpyllum* compared to the respective treatments in S-CC was observed (Figure 1, Figure 2 and Appendix A). Furthermore, under a light intensity of 174 µmol·m^−2^ s^−1^, an increase in the shoot length of *T. vulgaris* was observed in the control and P plants compared to their respective treatments in S-CC, and in *T. serpyllum* the increase was observed in all treatments (Figure 3 and Figure 4). Additionally, the light intensity of 600 µmol·m^−2^ s^−1^ caused an increase in shoot length under the P treatment in *T. vulgaris*, as well as under the TR and TR+P treatments in *T. serpyllum*, compared to the respective treatments in S-CC.

Regarding the food polymers, their significant effect on the increase in *T. serpyllum* shoot biomass as compared to the control was observed under the P treatment and a light intensity of 174 µmol·m^−2^ s^−1^, and their effect on the decrease in *T. vulgaris* shoot length as compared to all other treatments was observed under the P treatment in S-CC. This decrease was inhibited by the presence of *Trichoderma* under the TR+P treatment.

### 2.2. The Content of Plant Pigments in the Leaves of T. vulgaris and T. serpyllum

Regarding plant pigments, the results of two-way ANOVA showed that lighting intensity of 174 and 600 µmol·m^−2^ s^−1^ did not significantly influence Chl a and Chl b content in *T. vulgaris*, but in combination with soil treatment, the influence was significant (Table 1 and Appendix A). As compared to the control, the positive effect of soil treatment with TR, P, and TR+P on the Chl a and Chl b content in *T. vulgaris* plants growing under lighting of 600 µmol·m^−2^ s^−1^ was observed.

In *T. serpyllum*, both lighting, soil treatment, and a combination of these two factors, influenced Chl a and b contents (Appendix A). Under lighting conditions of 174 and 600 µmol·m^−2^ s^−1^, the content of Chl a and Chl b decreased in *T. serpyllum* control plants, as compared to S-CC (Table 2). Additionally, a decrease in the content of Chl a and Chl b was observed in the TR+P treatment under light intensity of 174 µmol·m^−2^ s^−1^, and in the P treatment, under 600 µmol·m^−2^ s^−1^ compared to the respective treatments in the S-CC. The Chl a content also decreased in other treatments under light intensity of 600 µmol·m^−2^ s^−1^.

Regarding carotenoids and anthocyanins in *T. vulgaris*, lighting and the combination of lighting with soil supplementation influenced their contents (Appendix A). An increase in the content of both pigments was observed under lighting conditions of 174 µmol·m^−2^ s^−1^ in control plants. Additionally, anthocyanins increased in *T. vulgaris* plants under P treatment. Under lighting conditions of 600 µmol·m^−2^ s^−1^, an increase in the content of both pigments was observed in all treatments, except for carotenoids in control plants, compared to the corresponding treatments in S-CC. In *T. serpyllum*, lighting significantly influenced anthocyanin contents, which decreased in the control and TR+P treatment under lighting conditions of 174 µmol·m^−2^ s^−1^ and 600 µmol·m^−2^ s^−1^ compared to the corresponding treatments in S-CC.

### 2.3. The Total Antioxidant Capacity and Total Phenolic, Flavonoid, and Terpenoid Contents

In the present studies, both lighting, soil treatment, and the combination of these two factors influenced total phenolic, flavonoid, and terpenoid contents and total antioxidant capacity in *T. vulgaris* and *T. serpyllum* plants (Appendix A).

In *T. vulgaris* and *T. serpyllum*, under light intensities of 174 and 600 µmol·m^−2^ s^−1^, an increase in the total phenolic content compared to the respective treatments in S-CC was observed (Figure 5 and Figure 6). Regarding *Trichoderma*, the use of microorganisms separately or in combination with polymers led to an increase in the phenolic content as compared to the control in *T. vulgaris* under lighting conditions of 174 µmol·m^−2^ s^−1^. *Trichoderma* treatment also increased the phenolic content in *T. serpyllum* grown in S-CC.

In our research, the content of flavonoids in plants grown in S-CC was much lower than in plants grown in conditions with additional lighting, both at 174 and 600 µmol·m^−2^ s^−1^ (Figure 7 and Figure 8). The only exceptions were the control plants and, in the case of *T. vulgaris*, the P plants grown under lighting conditions of 174 µmol·m^−2^ s^−1^. In addition, the simultaneous or separate application of *Trichoderma* with food polymers resulted in an increase in flavonoid content in *T. vulgaris* growing under a light intensity of 174 µmol·m^−2^ s^−1^ compared to the corresponding controls. Moreover, the use of microorganisms and polymers separately led to an increase in the flavonoid content of *T. serpyllum* growing under a light intensity of 174 µmol·m^−2^ s^−1^. Treatments of soil with *Trichoderma*, polymers, or their combination enhanced flavonoid content in all tested plants growing under lighting of 600 µmol·m^−2^ s^−1^, as compared to the respective controls. The highest flavonoid content of all results was observed in *T. serpyllum* treated with polymers and *Trichoderma*, grown under lighting of 600 µmol·m^−2^ s^−1^.

The total antioxidant capacity of *T. vulgaris* and *T. serpyllum* increased in parallel with the increase in phenolic content, including flavonoids, when additional lighting was used (Figure 9 and Figure 10), with only one exception: *T. vulgaris* P plants growing under a light intensity of 174 µmol·m^−2^ s^−1^. Moreover, the use of microorganisms separately or in combination with polymers led to an increase in the total antioxidant capacity in plants growing in the light intensity of 174 µmol·m^−2^ s^−1^ as compared do the control. In *T. serpyllum*, growing under a light intensity of 174 µmol·m^−2^ s^−1^, both treatments of soil with *Trichoderma* and polymers, separately, also increased the total antioxidant capacity.

The increases in terpenoid contents were recorded within individual lighting groups, which appear to be related to the presence of *Trichoderma* and/or food polymers in the soil. As compared to the S-CC conditions, significant increases in terpenoid contents were observed in *T. vulgaris* plants treated with *Trichoderma* or polymers growing under lighting of 174 µmol·m^−2^ s^−1^, *T. vulgaris* plants treated with polymers growing under lighting of 600 µmol·m^−2^ s^−1^, and *T. serpyllum* plants treated with polymers growing under lighting of 174 µmol·m^−2^ s^−1^ (Figure 11 and Figure 12). Additionally, as compared to the respective controls, in S-CC and 600 µmol·m^−2^ s^−1^ conditions, the increase in the terpenoid content was observed in *T. vulgaris* plants treated with *Trichoderma* and polymers. Regarding *T. serpyllum*, there was a noticeable increase in the content of terpenoids in P and TR+P plants treated with polymers separately or in combination with *Trichoderma*. The increase observed in TR+P plants was significantly greater compared to all other treatments.

### 2.4. Volatile Compounds Analysis by Using E-Nose

Eight volatile organic compounds (VOCs) were identified tentatively in the vapours of T. vulgaris and T. serpyllum (Table 3 and Table 4). The potentially identified compounds included four cyclic monoterpenes, α-pinene, myrcene, α-terpinene and γ-terpinene; one monoterpene cyclic ether, 1,8-cineole (eucalyptol); and three monoterpenoid phenols, thymol, carvacrol, and eugenol.

Thymol and carvacrol, as the major volatile monoterpenoid phenols, were identified directly on the basis of the retention time of the analytical standards. Table 3 and Table 4 show the relative content of volatile compounds expressed as a percentage of the total peak’s area. α-terpinene had the highest percentage share in all analysed samples, ranging from 16.78 to 43.03% for *T. vulgaris* and from 12.85 to 32.34% for *T. serpyllum*. The most important bioactive compound, thymol, occurred in amounts ranging from 1.70 to 5.31% for *T. vulgaris* and 2.56 to 6.61% for *T. serpyllum*. The lowest thymol content (1.7%) was recorded in the vapours of control *T. vulgaris* plants grown without additional lighting. A similarly low value was observed for control *T. serpyllum* plants grown without additional lighting. Furthermore, there was a tendency for the percentage of thymol in the vapours of plants grown with additional lighting of 174 µmol·m^−2^ s^−1^ to increase.

A considerable increase in the content of γ-terpinene was noticed in the case of *T. vulgaris* plants grown under additional lighting of both 174 µmol·m^−2^ s^−1^ and 600 µmol·m^−2^ s^−1^. A similar trend was observed in the case of *T. serpyllum* plants, with the relative content of γ-terpinene increasing 2–3 times compared to the content of this compound in the vapours of plants grown without additional lighting. Some interesting observations could be made concerning eugenol. The percentage of eugenol decreased under the influence of additional lighting. In some cases, the concentration of eugenol was reduced by up to seven times when compared to control plants grown without additional lighting.

Regarding the effect of *Trichoderma* and food polymers on thyme VOC contents in *T. vulgaris*, it seems that the microorganisms and polymers applied separately or together have a positive effect on thymol content in S-CC and CC600 plants as compared to the respective controls. Additionally, under the same conditions, the application of polymers without and with *Trichoderma* additionally positively affected eugenol content. The application of *Trichoderma* and food polymers resulted in even more pronounced increases in the contents of selected compounds in *T. serpyllum*. The microorganisms and polymers, whether applied separately or together, had a positive effect on the thymol and eugenol contents in S-CC, γ-terpinene in CC174, and the α-pinene contents in CC600 plants. Additionally, the application of polymers, with or without *Trichoderma*, positively affected the γ-terpinene contents in S-CC, as well as the α-pinene, myrcene, and thymol contents in CC174 plants.

## 3. Discussion

Plant growth and development are influenced by soil composition, including nutrient source and microorganism presence, as well as environmental conditions such as light, humidity, and temperature [35,36]. This study investigated the effect of lighting and soil supplementation with microorganisms classified as BCAs and a mixture of food polymers, including alginate, guar gum, and xanthan gum, on the growth and quality of *T. vulgaris* and *T. serpyllum* plants grown in semi-controlled conditions (S-CC) and controlled conditions with two lighting variants (174 and 600 µmol·m^−2^ s^−1^).

The results of our studies indicate a positive correlation between additional lighting and thyme growth. In particular, significant differences were observed in plant fresh biomass, with additional lighting increasing it several-fold. Furthermore, the supplementation of soil with food polymers, such as alginates and gums, led to greater biomass under certain lighting conditions depending on the thyme species. Although *Trichoderma* has been demonstrated to have a beneficial impact on the growth of different plants, including herbs, in various experimental systems [37,38], the strains used in this study did not significantly enhance the biomass or shoot length of thyme.

Chl a, Chl b, carotenoids, and anthocyanins are essential plant pigments that enable plants to function in the environment. They participate in photosynthesis, plant growth, development, and reproduction, as well as in defence responses against abiotic and biotic stresses. Pigments are commonly used as additives or supplements in the food industry, cosmetics, pharmaceuticals, and livestock feed, and they have a positive effect on human health [39,40].

In the present study, the contents of all pigments did not reveal any discernible patterns that could suggest a more significant impact of *Trichoderma* fungi or food polymers than lighting conditions on the pigment composition. However, research indicates that lighting affects plants differently depending on the species. In *T. vulgaris*, lighting has no effect or may even increase the content of pigments, particularly anthocyanins, which are products of secondary metabolism [41]. Anthocyanins are strong pigments belonging to flavonoids. Currently, it is believed that they perform protective functions in plants exposed to stress, including protection against excessive UV radiation and the formation of an internal optical filter that absorbs excess light [41,42]. In the present study, the production of these pigments increased in plants without a simultaneous decrease in Chl content, which may confirm their protective function.

Another finding relates to *T. serpyllum* plants. It seems that when they are treated separately with *Trichoderma* fungi or food polymers, the decrease in the content of all pigments is inhibited, which is particularly noticeable in TR and P plants grown under lighting conditions of 174 µmol·m^−2^ s^−1^. The reason for this response is currently unknown. However, it can be assumed that treating plants with BCAs and organic substances such as alginates and gums under certain lighting conditions may help to protect plant pigments or influence their production. For example, *Trichoderma rifaii* and *Pseudomonas* spp. have been shown to have a positive effect on Chl and anthocyanin contents in the medicinal herb roselle (*Hibiscus sabdariffa* L.) [43], while organic fertilisers and *Trichoderma harzianum* significantly increased the Chl index in black cumin (*Nigella sativa* L.) [44].

Among environmental factors, light plays an important role in determining the desirable physical and biochemical characteristics of plants. Excess light can constitute an abiotic stress for plants and influence secondary metabolism in plants. Secondary metabolism refers to the biochemical pathways that produce compounds not directly involved in the growth, development, or reproduction of plants but often playing crucial roles in defence against various stresses. The intensity and quality of light control various plant responses, including the production of secondary metabolites such as polyphenolic compounds, especially flavonoids, and terpenoids [45].

The quantity and quality of secondary metabolites produced by plants, including those with health-promoting properties, can also be significantly influenced by the composition of the soil and the microbiological environment. This influence may vary depending on the specific characteristics of the environment [46]. In the present study, total phenolics, flavonoids, and terpenoids were considered the main groups of secondary metabolites determining the antioxidant capacity and health-promoting properties of thyme. In the present studies, a strong effect of additional lighting on the increase in the phenolic content was confirmed. In addition, the lighting reversed the decrease in phenolic content observed in a few cases. For example, the application of polymers, either separately or in combination with *Trichoderma* in S-CC, resulted in lower concentrations of phenolics compared to the control in the case of *T. vulgaris*. However, under light intensities of 174 and 600 µmol·m^−2^ s^−1^, this effect was eliminated, suggesting that soil enrichment with microorganisms and food polymers is most effective when combined with additional lighting.

The sources of phenolic compounds were not analysed in our studies. However, it is reasonable to assume that the increase in their levels may be due to the increased activity of phenylalanine ammonia lyase (PAL). The PAL enzyme plays a crucial role in the phenylpropanoid pathway, where it catalyses the deamination of L-phenylalanine to trans-cinnamic acid. This metabolic pathway is essential for the synthesis of various secondary metabolites, including flavonoids, lignin and phytoalexins, which are important for plant structural integrity and defence against stress [47]. In response to light exposure, particularly UV-B light, the activation of genes associated with secondary metabolite production can be observed. There is evidence that UV exposure leads to the accumulation of phenolics and the de novo synthesis of PAL proteins [48]. Moreover, qualitative and quantitative changes in the profile of polyphenolic compounds are also observed [49,50,51]. Excess light and UV radiation are important factors that enhance the production of plant flavonoids belonging to a class of secondary metabolites with multiple functions, including pigmentation, photoprotection, and oxidative protection [41,52]. In addition to their typical antioxidant properties, flavonoids that bind to membrane proteins and phospholipids can, for example, stabilise their structure by reducing their fluidity, which in turn hinders the diffusion of free radicals and limits the peroxidation of membrane lipids [53]. Plants accumulate flavonoids in the vacuoles of epidermal cells to protect the underlying tissues from the damaging effects of UV radiation. This important role of flavonoids has also been confirmed by research conducted by Ryan et al. [54] on *Arabidopsis* mutants deficient in an enzyme necessary for the synthesis of flavonoid compounds, which showed significant hypersensitivity to UV-B radiation.

The results of the present studies indicate that supplementing soil with microorganisms and food polymers has a positive effect on flavonoid synthesis in plants exposed to additional lighting. The beneficial impact of microorganisms on flavonoid synthesis has been reported in other studies. For example, Şesan et al. [55] demonstrated that treating *Passiflora caerulea* L. with a *Trichoderma* consortium, consisting of two strains, *T. asperellum* T36b and *T. harzianum* Td50b, improved plant yield and quality. This improvement was associated with the accumulation of polyphenols and flavonoids, as well as increased antioxidant activity. The increased flavonoid content was also observed in pear-jujube (*Zizyphus jujuba* Mill.cv.), planted in soil containing organic fertilisers. This resulted in a significant improvement in the physiological state and fruit quality of the plant [56].

Similar to the changes in phenolic compounds, the application of *Trichoderma* and food polymers, either simultaneously or separately, resulted in an increase in flavonoid content in *T. vulgaris* or *T. serpyllum*. This trend was observed in plants growing under a light intensity of 174 µmol·m^−2^ s^−1^ compared to the corresponding controls. The results suggest that phenolic compounds, including flavonoids, contribute to the total antioxidant capacity of *T. vulgaris* and *T. serpyllum*.

Terpenoids, also known as isoprenoids, are a large family of plant metabolites. They are found in many herbal plants, and several terpenoids have been shown to be available for pharmaceutical applications [57]. In plants, terpenoids are primarily used for abiotic and biotic interactions and defence, while humans use them mainly in the pharmaceutical, food, and chemical industries [58].

The obtained results indicate that the use of *Trichoderma* and food polymers, either separately or in combination, may positively affect the synthesis of terpenoids. However, the specific changes depend on the species of thyme. Although the mechanisms of terpenoid synthesis induction by BCAs and organic fertilisers added to the soil are not well understood, it can be assumed that a positive induction of methylerythritol 1-phosphate, lipid, and shikimate pathways occurs in the tested plants. Such effects have been observed, for example, in *Trichoderma* spp.-treated olive (*Olea europaea* L.) trees [59] and organic fertiliser-treated rosemary (*Rosmarinus officinalis*) plants [60].

The last part of the research concerned the detection of volatile compounds in the vapours of *T. vulgaris* and *T. serpyllum*. All identified volatile compounds have previously been reported as constituents of essential oils isolated from the genus *Thymus* [13,61]. Among the detected volatile compounds, special attention may be paid to γ-terpinene, whose content increased under different conditions, including additional lighting, and was positively influenced by *Trichoderma* and food polymers. These results indicate that food polymers and, to a lesser extent, *Trichoderma* have a positive effect on the synthesis of volatile compounds with health-promoting properties. This is consistent with other studies that have shown a positive influence of natural fertilisation on thymol generation by *T. vulgaris* [62,63].

The accumulation of secondary metabolites has been identified as an adaptive strategy employed by plants in response to stress. The observed changes in phenolic compounds and terpene contents, in parallel with the increased content of volatile compounds in the vapours of *T. vulgaris* and *T. serpyllum*, may be considered indicative of defence responses in the tested plants. These responses could potentially be activated by additional lighting. Lee et al. [64] demonstrated that rice seedlings exposed to UV-B radiation emitted a variety of monoterpenes in a time-dependent manner. The mixtures comprised the following components: limonene, sabinene, myrcene, α-terpinene, β-ocimene, γ-terpinene, and α-terpinolene. In a different context, the application of supplementary lighting of various intensities, as well as UV radiation, is commonly used for the production of secondary metabolites in vitro [20]. Furthermore, the application of supplementary lighting has been observed to stimulate the synthesis of a range of secondary metabolites, including terpenes, with potential health effects in *Cannabis sativa* L. [65]. The enhanced emission of volatile compounds was observed also during the defence and tolerance response of plants to other stresses. For example, thymol improved the salinity tolerance of *Nicotiana tabacum* [66].

Some interesting observations can be made concerning eugenol. Despite its concentration being positively influenced by the application of polymers without and with Trichoderma, in general, the percentage of eugenol decreased under the influence of additional lighting. This finding is difficult to explain within the scope of the present study as there is limited knowledge regarding the potential impact of eugenol on plant stress resistance. However, initial studies suggest that eugenol may contribute to abiotic stress tolerance. Zhao et al. [67] provided the first evidence that volatile eugenol can be absorbed by tea plants (*Camellia sinensis*) and metabolised to glycosides, enhancing the plants’ ability to tolerate cold and drought conditions. Other authors demonstrated that eugenol is a crucial factor in regulating plant resistance physiology [68] In this study, tobacco seedlings were exposed to a NaCl solution with or without eugenol. The growth of the seedlings was inhibited by salt stress, but this effect was effectively mitigated in a dose-dependent manner by eugenol supplementation.

In conclusion, the present study demonstrated that the growth of *T. vulgaris* and *T. serpyllum* seedlings was significantly accelerated under the influence of additional lighting, resulting in an increase in shoot length and fresh biomass. Furthermore, an increase in carotenoids was observed in *T. vulgaris* and *T. serpyllum* seedlings. With regard to secondary metabolites, the most pronounced changes were observed in total antioxidant capacity and flavonoid content, which increased significantly under the influence of additional lighting, as well as the influence of *Trichoderma*, food polymers, or both, depending on the thyme species. The simultaneous or separate application of *Trichoderma* and food polymers resulted in an increase in flavonoid content in the leaves of both species. The increase in terpenoid content under supplemental light appears to be related to the presence of *Trichoderma* spores as well as food polymers added to the soil. However, the nature of these changes depends on the thyme species. Eight volatile compounds (VOCs) were tentatively identified in the vapours of *T. vulgaris* and *T. serpyllum*: α-pinene, myrcene, α-terpinene, γ-terpinene; 1,8-cineole (eucalyptol), thymol, carvacrol, and eugenol. There was a tendency for the percentage of thymol and γ-terpinene to increase under supplemental lighting. The results also demonstrate a positive effect of food polymers and, to a lesser extent, *Trichoderma* fungi, on the synthesis of VOCs with health-promoting properties. The effect of *Trichoderma* and food polymers on individual VOCs was positive in some cases for thymol and γ-terpinene. On the other hand, the percentage of eugenol in the vapours of both species decreased under the influence of supplementary lighting. In conclusion, it is possible that the addition of food polymers, including alginates, gums, and *Trichoderma*, separately rather than in combination, may positively affect the synthesis of valuable metabolites of thyme. The results of this study indicate that it is challenging to definitively determine which species of thyme exhibited greater resistance to the stress associated with intense lighting. The additional lighting intensity of 174 µmol·m^−2^·s^−1^ appears to be sufficient to stimulate the growth and development of both tested herb species and to stimulate the synthesis of the desired secondary metabolites. The results obtained encourage further research to characterise the effects of lighting in combination with soil supplementation on the metabolism of thyme plants and other economically important herbs. In planning future research, we would like to focus on developing sustainable cultivation practices that not only improve herb quality and enhance bioactive metabolite contents but also allow the use of biofertilisers derived from the food industry in the form of unused waste. Further plans are embedded in the current trend of studies investigating the potential for the use of organic and environmentally friendly soil treatments and renewable energy sources for lighting in order to align herb cultivation with global sustainability goals.

## 4. Materials and Methods

### 4.1. Trichoderma Inoculum Preparation

*T. virens* TRS 106 and *T. atroviride* TRS 25 were obtained from the bank collection of the Department of Microbiology and Rhizosphere, Institute of Horticulture—National Research Institute in Skierniewice (Skierniewice, Poland). The morphological identification and molecular classification of TRS 106 and TRS 25 were described previously and deposited in the NCBI GenBank [69,70,71]. TRS 106 and TRS 25 isolates were previously characterised as nonpathogenic microorganisms able to promote crop plant growth and induce resistance against different pathogens [70,71].

Before being used for plant treatment, TRS 106 and TRS 25 were grown on Malt Extract Agar medium in an Incubator Incucell for 10 days at 25 °C and exposed every 24 h to daylight for 20 min. Such conditions were the most optimal for the efficient sporulation of TRS 106 and TRS 25. To obtain Trichoderma inoculum, the spores of the fungus from one Petri plate were washed off the surface with 10 mL of 0.85% NaCl solution.

### 4.2. Plant Material, Growth Conditions, and Treatment of Plants with Trichoderma and Polymeric Substances

The experiment was carried out under semi-controlled (S-CC) and controlled conditions (CCs). Seeds from *Thymus vulgaris* “Sunshine” and *Thymus serpyllum* “Pinklilac” were sown into multi-pots filled with a peat substrate in a growing chamber. The seeds were carefully selected and provided by W. Legutko Producer under the supervision of the Voivodeship Plant Health and Seed Inspection Service. The growing conditions were optimal for the seeds to thrive. After three weeks, the seedlings (four seedlings per pot) of each species were randomly selected and planted in CCs in growing chambers under two lighting conditions, (i) a chamber with lighting of 174 µmol·m^−2^ s^−1^ photon flux density (lamps type 36W, Philips TDL 36/84) or (ii) a chamber with lighting of 600 µmol·m^−2^ s^−1^ photon flux density (lamps type NEONICA GROWY LED TOP PLUS 240), or into the S-CCs growing chamber. Two different intensities of lighting were selected: one to support photosynthesis and growth and the other to support secondary metabolism. It is worth noting that in some plants, the latter intensity is close to the level that induces stress responses.

In the CCs, the plants were grown at 25/20 °C with a 14/10 h day/night photoperiod at 70% relative humidity. Regarding the plants grown in the S-CC growing room, the changing parameters were recorded (Table 5).

In each experimental condition, four experimental groups of plants were tested:C—control plants;TR—plants grown in a soil supplemented with *T. virens* TRS 106 spores (TRS 106) and *T. atroviride* TRS 25 spores (TRS 25);P—plants grown in a soil supplemented with polymeric substances;TR+P—plants grown in the soil supplemented with TRS 106/25 spores, and polymeric substances.

For all plants, the base soil comprised a peat-based substrate (80–120 mg N, 60–80 mg P, 160–220 mg K, 70–120 mg Mg, and 1500–2000 mg Ca/L at pH 5.5–6.0) mixed with perlite at a ratio of 1:0.25 (*v*:*v*).

Regarding P plants, polymer substances in the form of a water solution (6 µM sodium alginate, 1 µM guar gum, and 1 µM xanthan gum) were added to the soil described above. This was done for three-week-old seedlings transplanted from the peat substrate into pots. The concentrations of food polymers were chosen based on average values used in food products. Regarding TR plants, *Trichoderma* spores were introduced into the soil described above. This was done for four-week-old seedlings transplanted from peat substrate into pots, aiming to achieve a spore density of 10^3^ per 1 g of substrate for both *T. virens* TRS 106 and *T. atroviride* TRS 25. Finally, 9 experimental treatments were obtained for each thyme variety (Figure 13).

### 4.3. T. vulgaris and T. serpyllum Leaves Extraction for Total Phenolic, Flavonoids Content, and Total Antioxidant Capacity

An amount of 0.25 g of fresh plant tissue was ground in 5 mL of 40% ethanol solution using a mortar and quantitatively transferred to a plastic centrifuge tube. The solid/solvent ratio was 1/20 (*w*/*v*). The solution was then centrifuged at 6000 rpm (4800 G) for 10 min using a Centurion Scientific K2015R centrifuge (Stoughton, UK). Supernatants were poured into new tubes and stored in a freezer at −40 °C until analyses.

Based on our previous experience, in our current research, we decided to carry out extraction in 40% aqueous ethanol solution as a solvent. The results of our earlier studies [72] showed that the use of higher ethanol concentrations did not result in higher extraction of polyphenols. Moreover, ethanol at a concentration of 80% caused a decrease in extraction efficiency. Perhaps the decrease in extraction efficiency at high ethanol concentrations is caused by the denaturation of proteins, thus blocking the release of biologically active compounds from plant cells. The literature data seem to confirm our assumptions. In extracts from *Lavendula* stoechas (also *Lamiaceae* family), the maximum extraction efficiency and the highest overall polyphenol concentration were achieved using a 40% ethanol solution [73]. Similarly, using ethanol at a concentration of 50% was optimal for the extraction of polyphenols from *T. serpyllum*. Furthermore, it was observed that the extraction duration had no significant impact on the polyphenol concentration [74].

### 4.4. Determination of Total Phenolic Content

The content of phenolic compounds was determined using the Folin–Ciocalteu assay originally described by Singleton and Rossi [75]. An amount of 0.1 mL of each extract was added to 3.8 mL of distilled water and 0.1 mL of the Folin–Ciocalteu reagent. The mixture was incubated in the dark for 3 min. Then, 1 mL of 10% (*w*/*v*) Na_2_CO_3_ solution was added to the mixture and incubated for 60 min at room temperature in the dark. Absorbance was measured at 765 nm using a UV-1800 spectrophotometer (Shimadzu, Tokyo, Japan). Experiments were performed in 6 replicates. A standard curve was constructed using gallic acid solutions ranging from 0.05 to 0.4 mg/mL. Concentrations of total phenolic compounds were expressed as mg gallic acid equivalent per 1 g of fresh weight (mg GAE/g).

### 4.5. Determination of Flavonoid Content

The flavonoid content of plant tissues was determined by the method described by Chang et al. [76] with some modifications. An amount of 0.5 mL of each extract was added to 1.5 mL of 80% (*w*/*v*) ethanol solution, 0.1 mL of 10% (*w*/*v*) AlCl_3_·6H_2_O, 0.1 mL of 1 M CH_3_COONa, and 2.8 mL of distilled water. The mixture was incubated in the dark at room temperature for 40 min. Absorbance was measured at 415 nm using a UV-1800 spectrophotometer (Shimadzu, Tokyo, Japan). Experiments were performed in 6 replicates. The standard curve was prepared using quercetin solutions ranging from 0.01 to 0.1 mg/mL. The final flavonoid content was expressed as mg quercetin equivalent per 1 g of fresh weight (mg quercetin/g).

### 4.6. Determination of Total Antioxidant Capacity

Total antioxidant capacity was determined using the FRAP method (Ferric Reducing Antioxidant Power) as described by Vignoli et al. [77] with modifications. Each extract (0.1 mL) was added to 4 mL of FRAP solution, which was prepared by adding 2.5 mL of a 10 mM TPTZ solution (2,4,6-tripyridyl-1,3,5-triazine) in 40 mM HCl, 2.5 mL of a 20 mM FeCl_3_·6H_2_O solution, and 25 mL of 0.3 mM acetate buffer at pH 3.6 to a 100 mL flask. The mixture was filled with distilled water up to 100 mL, closed tightly, and mixed thoroughly. The extracts were incubated with FRAP solution for 30 min in the dark. The absorbance was measured at 593 nm using a UV-1800 spectrophotometer (Shimadzu, Tokyo, Japan). All analyses were performed in 6 replicates. Trolox solutions with concentrations ranging from 40 to 4000 μM were used to prepare a standard curve. The total antioxidant capacity was expressed as μmol TE per 1 g of fresh weight.

### 4.7. Determination of Chlorophyll a and b, Anthocyanin, and Carotenoid Contents

The content of photosynthetic pigments in plant tissues was determined by using the method described by Hiscox and Israelstam [78] with some modifications. Five circles with a diameter of 9 mm were cut out from freshly collected leaves from each plant. The cut-out circles were then placed in test tubes and 5 mL of DMSO was added. The prepared samples were incubated in a water bath at 65 °C for 30 min. Immediately after this time, the absorbance (534 nm, 643 nm, 661 nm, 470 nm) was measured against the pure DMSO reagent. The concentrations of individual pigments were calculated according to Arnon’s formulae (Equations (1)–(4)) and expressed in mg/g of fresh weight.
Chlorophyll a [g/L] = 0.01261∙A_661_ − 0.001023∙A_534_ − 0.0022∙A_643_(1)
Chlorophyll b [g/L] = 0.02255∙A_643_ − 0.00439∙A_534_ − 0.004488∙A_661_(2)
Anthocyanins [g/L] = 0.082∙A_534_ − 0.00687∙A_643_ − 0.002423∙A_661_(3)
Carotenoids [g/L] = (A_470_ − 17.1∙(Chl a + Chl b) − 9.479∙Anthocyanins)/119.26(4)

### 4.8. Determination of Terpenoid Content

The content of terpenoids in plant tissues was determined by using the method described by Ghorai et al. [79] with some modifications. An amount of 0.25 g of fresh plant tissue was ground in 2.5 mL of 40% ethanol solution using a mortar and quantitatively transferred to a plastic centrifuge tube. The solid/solvent ratio was 1/20 (*w*/*v*). The solution was then centrifuged at 6000 rpm (4800 G) for 15 min. Supernatants were poured into new tubes and stored in a freezer at −40 °C until analyses. Each extract (0.4 mL) was added to 3 mL of chloroform. The tubes were placed in ice for 3 min to rest, and to each of them, 0.2 mL of sulfuric acid (H_2_SO_4_) was added. The sample mixtures were mixed thoroughly and left in a freezer at −40 °C for 15 min to rest. The absorbance of the separated alcoholic layer containing terpenoids was measured at 538 nm using a UV-1800 spectrophotometer (Shimadzu, Tokyo, Japan). All analyses were performed in 6 replicates. Linalool solution with concentrations ranging from 100 mg/200 µL to 1 mg/200 µL was used to prepare a standard curve. The terpenoid content was expressed in mg per 1 g of fresh weight.

### 4.9. Electronic Nose Analysis

Volatile constituents were analysed using an electronic nose (E-nose) following the method described by Rottiers et al. [80] with some modifications. The experiment was performed using a Heracles II electronic nose (Alpha MOS, Toulouse, France) equipped with an HS-100 autosampler, a sensor array unit, and a non-polar column (MXT5: 5% diphenyl, 95% methylpolysiloxane, 10 m length and 180 lm diameter). An amount of 0.5 g of fresh leaves was placed in 20 mL screw vials with polytetrafluoroethylene-silicone septa and sealed with a magnetic cap. The vials were then incubated in a shaker oven at 50 °C and shaken at 500 rpm for 20 min. After that, 1000 µL of the headspace was sampled using a syringe and injected into the gas chromatograph with two flame ionisation detectors. At the start of the process, the temperature was raised from 50 °C to 250 °C at a rate of 3 °C/s and held for 21 s. The total separation time was 100 s. The apparatus was calibrated using a solution of alkanes, ranging from n-hexane to n-hexadecane. The Kovats retention indices and volatile compounds were identified using AroChemBase V6 software, version 6.0-4.3.0 (Alpha MOS, Toulouse, France) based on the retention times of n-alkanes. Each sample was measured three times. Data acquisition and subsequent analysis were performed using Alphasoft 14.2 and AroChembase (Alpha MOS, Toulouse, France) software. Quantitative analysis was based on the percentage of each peak’s area (relative area) in relation to all the peak areas of a given sample. The percentage composition of volatile compounds was calculated based on this information. Thymol and carvacrol, which are the major volatile monoterpenoid phenols, were identified using the retention time of the analytical standards (Thymol, CRM40188, Sigma Aldrich, USA and Carvacrol, 42632, Sigma Aldrich, St. Louis, MO, USA).

### 4.10. Statistical Analysis

The results of the biomass and length of shoots of thyme plants are the mean of 12 independent replications ± standard deviations (±SD), and the results of the biochemical analyses are the mean of 6 independent replications ± standard deviations (±SD). The effect of the lighting conditions and soil treatments and their interactions were checked with two-way ANOVA. The significant differences among the means were estimated through Duncan’s post hoc test. For all statistical analyses, *p* < 0.05 was considered statistically significant. All statistical evaluations were conducted using Statistica 13.0 software (StatSoft Inc., Palo Alto, CA, USA).

## Figures and Tables

**Figure 1 ijms-25-04846-f001:**
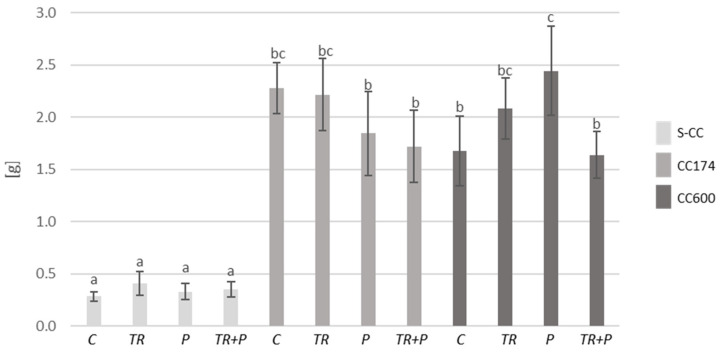
Shoot biomass (FW) of *T. vulgaris*. The values represent the means + SE from *n* = 12. Statistical analysis of variance (two-way ANOVA, *p* < 0.05, Duncan multiple range post hoc test) was performed. Different letters a, b and c indicate that samples are significantly different. The letter “a” marks the lowest value. Higher values are marked with consecutive letters of the alphabet. Bars that share the same letter within the group are not significantly different. Abbreviations: S-CC, semi-controlled conditions; CC174, controlled conditions with lighting 174 µmol·m^−2^ s^−1^; CC600, controlled conditions with lighting 600 µmol·m^−2^ s^−1^; C, control plants; TR, plants grown in the soil supplemented with *Trichoderma*; P, plants grown in the soil supplemented with food polymers; TR+P, plants grown in the soil supplemented with *Trichoderma* and food polymers.

**Figure 2 ijms-25-04846-f002:**
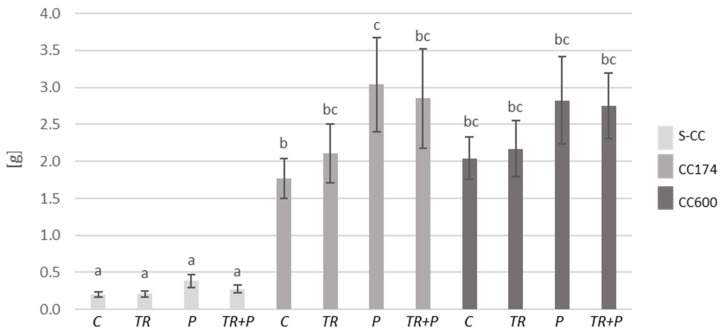
Shoot biomass (FW) of *T. serpyllum*. The values represent the means + SE from *n* = 12. Statistical analysis of variance (two-way ANOVA, *p* < 0.05, Duncan multiple range post hoc test) was performed. Different letters a, b and c indicate that samples are significantly different. The letter “a” marks the lowest value. Higher values are marked with consecutive letters of the alphabet. Bars that share the same letter within the group are not significantly different. Abbreviations as in Figure 1.

**Figure 3 ijms-25-04846-f003:**
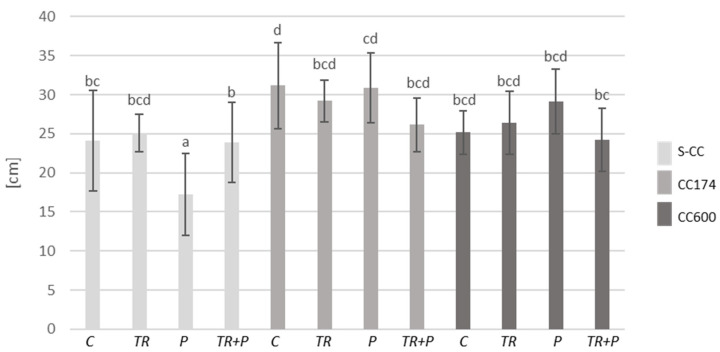
Length of shoots of *T. vulgaris*. The values represent the means + SE from *n* = 12. Statistical analysis of variance (two-way ANOVA, *p* < 0.05, Duncan multiple range post hoc test) was performed. Different letters a, b, c and d indicate that samples are significantly different. The letter “a” marks the lowest value. Higher values are marked with consecutive letters of the alphabet. Bars that share the same letter within the group are not significantly different. Abbreviations as in Figure 1.

**Figure 4 ijms-25-04846-f004:**
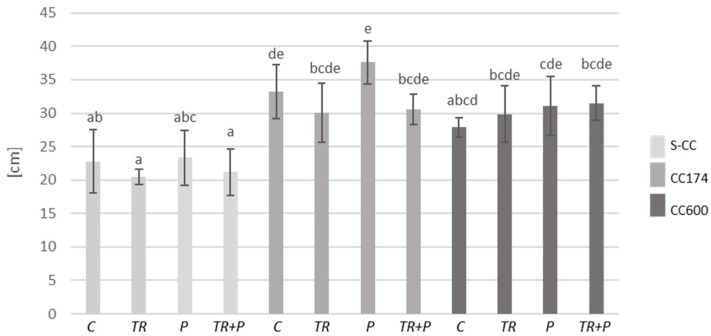
Length of shoots of *T. serpyllum*. The values represent the means + SE from *n* = 12. Statistical analysis of variance (two-way ANOVA, *p* < 0.05, Duncan multiple range post hoc test) was performed. Different letters a, b, c, d and e indicate that samples are significantly different. The letter “a” marks the lowest value. Higher values are marked with consecutive letters of the alphabet. Bars that share the same letter within the group are not significantly different. Abbreviations as in Figure 1.

**Figure 5 ijms-25-04846-f005:**
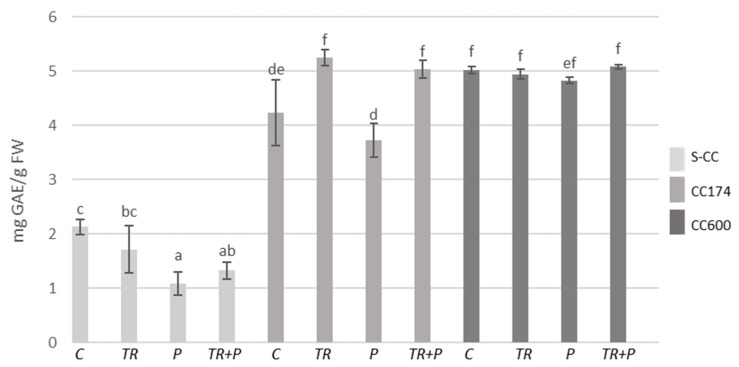
Total phenolic contents in *T. vulgaris*. Results are expressed in mg of GAE equivalent per 1 g of FW. The values represent the means + SE from *n* = 6. Statistical analysis of variance (two-way ANOVA, *p* < 0.05, Duncan multiple range post hoc test) was performed. Different letters a, b, c, d, e, and f indicate that samples are significantly different. The letter “a” marks the lowest value. Higher values are marked with consecutive letters of the alphabet. Bars that share the same letter within the group are not significantly different. Abbreviations as in Figure 1.

**Figure 6 ijms-25-04846-f006:**
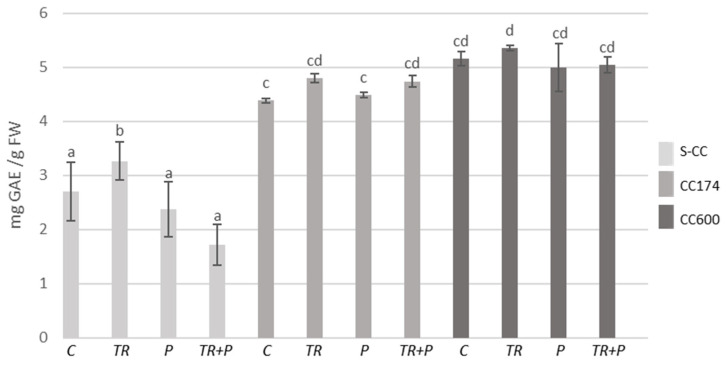
Total phenolic contents in *T. serpyllum*. Results are expressed in mg of GAE equivalent per 1 g of FW. The values represent the means + SE from *n* = 6. Statistical analysis of variance (two-way ANOVA, *p* < 0.05, Duncan multiple range post hoc test) was performed. Different letters a, b, c, and d indicate that samples are significantly different. The letter “a” marks the lowest value. Higher values are marked with consecutive letters of the alphabet. Bars that share the same letter within the group are not significantly different. Abbreviations as in Figure 1.

**Figure 7 ijms-25-04846-f007:**
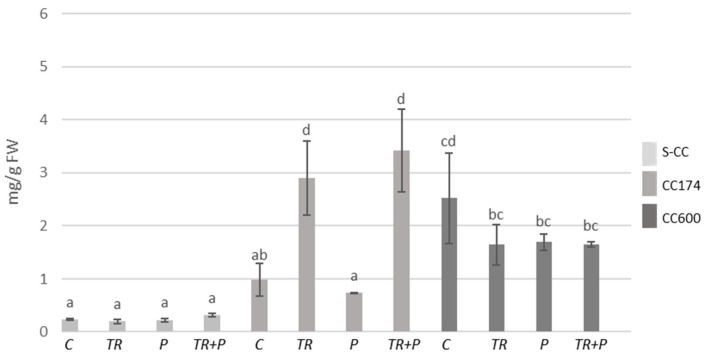
Flavonoid contents in *T. vulgaris*. Results are expressed in mg of quercetin per 1 g of FW. The values represent the means + SE from *n* = 6. Statistical analysis of variance (two-way ANOVA, *p* < 0.05, Duncan multiple range post hoc test) was performed. Different letters a, b, c, and d indicate that samples are significantly different. The letter “a” marks the lowest value. Higher values are marked with consecutive letters of the alphabet. Bars that share the same letter within the group are not significantly different. Abbreviations as in Figure 1.

**Figure 8 ijms-25-04846-f008:**
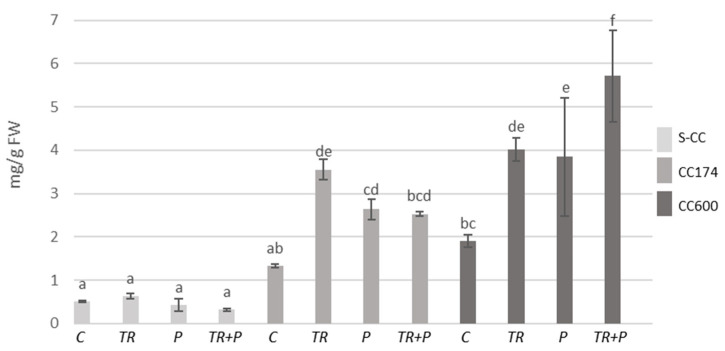
Flavonoid contents in *T. serpyllum*. Results are expressed in mg of quercetin per 1 g of FW. The values represent the means + SE from *n* = 6. Statistical analysis of variance (two-way ANOVA, *p* < 0.05, Duncan multiple range post hoc test) was performed. Different letters a, b, c, d, e, and f indicate that samples are significantly different. The letter “a” marks the lowest value. Higher values are marked with consecutive letters of the alphabet. Bars that share the same letter within the group are not significantly different. Abbreviations as in Figure 1.

**Figure 9 ijms-25-04846-f009:**
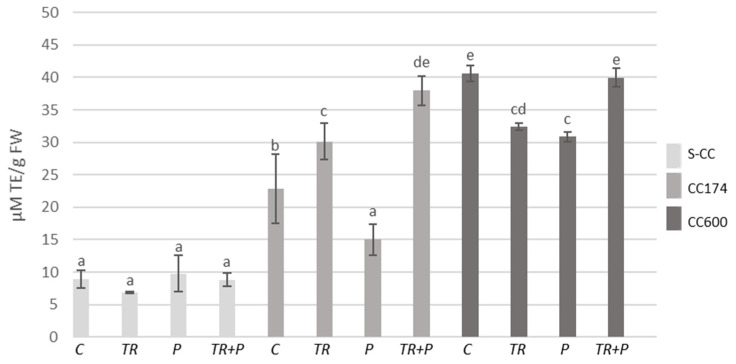
Total antioxidant capacity of *T. vulgaris*. Results are expressed in µM of Trolox equivalent per 1 g of FW. The values represent the means + SE from *n* = 6. Statistical analysis of variance (two-way ANOVA, *p* < 0.05, Duncan multiple range post hoc test) was performed. Different letters a, b, c, d, and e indicate that samples are significantly different. The letter “a” marks the lowest value. Higher values are marked with consecutive letters of the alphabet. Bars that share the same letter within the group are not significantly different. Abbreviations as in Figure 1.

**Figure 10 ijms-25-04846-f010:**
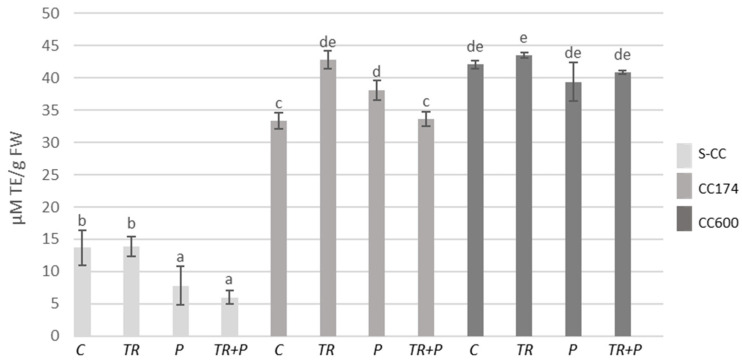
Total antioxidant capacity of *T. serpyllum*. Results are expressed in µM of Trolox equivalent per 1 g of FW. The values represent the means + SE from *n* = 6. Statistical analysis of variance (two-way ANOVA, *p* < 0.05, Duncan multiple range post hoc test) was performed. Different letters a, b, c, d and e indicate that samples are significantly different. The letter “a” marks the lowest value. Higher values are marked with consecutive letters of the alphabet. Bars that share the same letter within the group are not significantly different. Abbreviations as in Figure 1.

**Figure 11 ijms-25-04846-f011:**
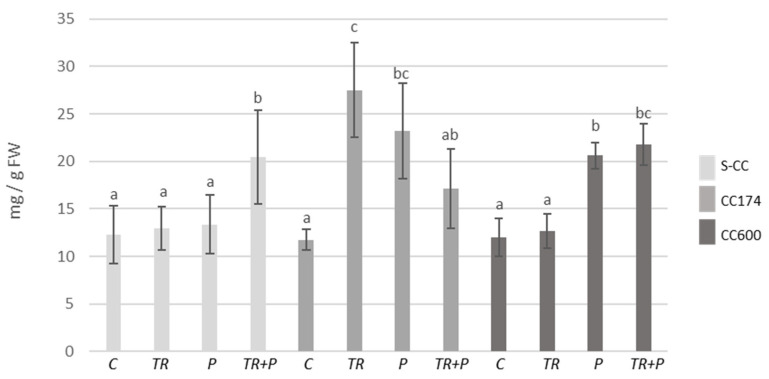
Terpenoid contents in *T. vulgaris*. Results are expressed in mg of linalool equivalent per 1 g of FW. The values represent the means + SE from *n* = 6. Statistical analysis of variance (two-way ANOVA, *p* < 0.05, Duncan multiple range post hoc test) was performed. Different letters a, b, and c indicate that samples are significantly different. The letter “a” marks the lowest value. Higher values are marked with consecutive letters of the alphabet. Bars that share the same letter within the group are not significantly different. Abbreviations as in Figure 1.

**Figure 12 ijms-25-04846-f012:**
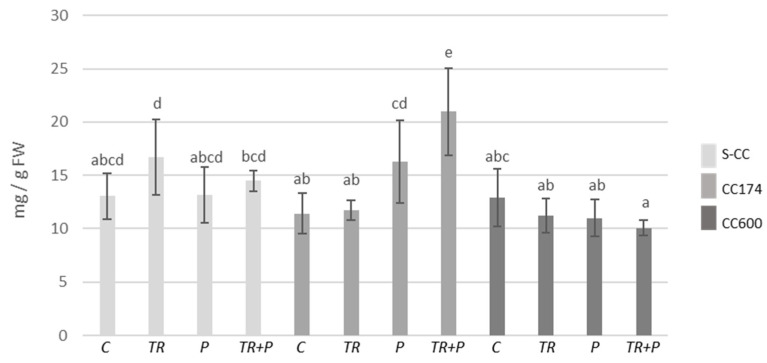
Terpenoid contents in *T. serpyllum*. Results are expressed in mg of linalool equivalent per 1 g of FW. The values represent the means + SE from *n* = 6. Statistical analysis of variance (two-way ANOVA, *p* < 0.05, Duncan multiple range post hoc test) was performed. Different letters a, b, c, d, and e indicate that samples are significantly different. The letter “a” marks the lowest value. Higher values are marked with consecutive letters of the alphabet. Bars that share the same letter within the group are not significantly different. Abbreviations as in Figure 1.

**Figure 13 ijms-25-04846-f013:**
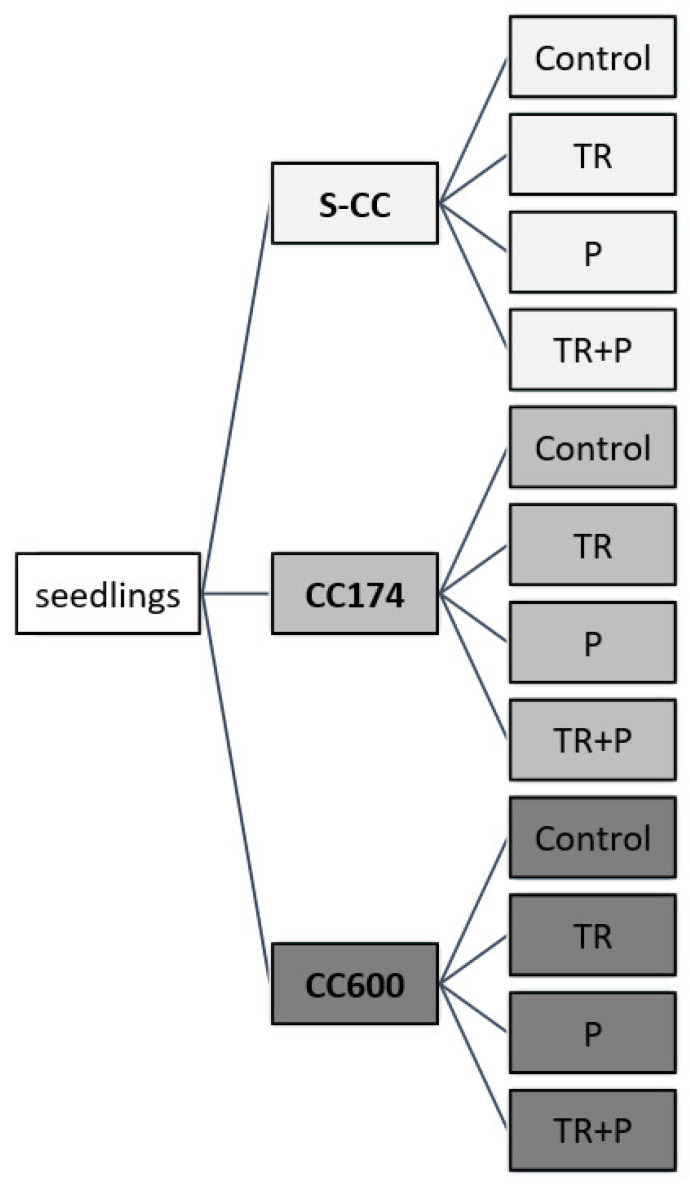
Study design. Abbreviations: S-CC, semi-controlled conditions; CC174, controlled conditions with lighting 174 µmol·m^−2^ s^−1^; CC600, controlled conditions with lighting 600 µmol·m^−2^ s^−1^; TR, plants grown in a medium containing *Trichoderma fungi*; P, plants grown in a medium containing food polymers; TR+P, plants grown in a medium containing both *Trichoderma* and food polymers.

**Table 1 ijms-25-04846-t001:** The content of plant pigments in the leaves of *T. vulgaris*. The values represent the means + SE from *n* = 6. Statistical analysis of variance (two-way ANOVA, *p* < 0.05, Duncan multiple range post hoc test) was performed. Different letters a, b, c, d, e, and f indicate that samples are significantly different. The letter “a” marks the lowest value. Higher values are marked with consecutive letters of the alphabet. Bars that share the same letter within the group are not significantly different. Abbreviations as in Figure 1.

Lighting Conditions	Plant/Treatment	Plant Pigments [mg g^−1^ FW]			
		Chlorophyll a	Chlorophyll b	Carotenoids	Anthocyanins
S-CC	C	1.141 ± 0.343 abcd	0.396 ± 0.125 bc	0.621 ± 0.176 a	0.548 ± 0.182 a
	TR	1.341 ± 0.290 cdef	0.325 ± 0.035 ab	0.589 ± 0.111 a	0.677 ± 0.104 ab
	P	1.410 ± 0.065 def	0.407 ± 0.025 bc	0.625 ± 0.019 ab	0.579 ± 0.016 a
	TR+P	1.254 ± 0.149 cdef	0.345 ± 0.080 ab	0.564 ± 0.017 a	0.707 ± 0.103 ab
CC174	C	1.185 ± 0.283 cde	0.428 ± 0.106 bc	0.930 ± 0.128 cd	1.161 ± 0.041 cd
	TR	1.429 ± 0.085 def	0.351 ± 0.025 abc	0.815 ± 0.064 abcd	0.922 ± 0.074 bc
	P	1.041 ± 0.108 abc	0.256 ± 0.045 a	0.691 ± 0.168 abc	0.857 ± 0.252 b
	TR+P	1.007 ± 0.219 abc	0.484 ± 0.147 c	0.733 ± 0.068 abcd	0.836 ± 0.191 ab
CC600	C	0.795 ± 0.126 a	0.255 ± 0.026 a	0.744 ± 0.106 abcd	1.114 ± 0.050 cd
	TR	1.590 ± 0.040 f	0.456 ± 0.113 bc	1.007 ± 0.172 d	1.132 ± 0.161 cd
	P	1.444 ± 0.204 def	0.392 ± 0.062 bc	0.984 ± 0.086 cd	1.358 ± 0.142 d
	TR+P	1.007 ± 0.219 ef	0.397 ± 0.036 bc	0.918 ± 0.037 bcd	1.125 ± 0.128 cd

**Table 2 ijms-25-04846-t002:** The content of plant pigments in the leaves of *T. serpyllum*. The values represent the means + SE from *n* = 6. Statistical analysis of variance (two-way ANOVA, *p* < 0.05, Duncan multiple range post hoc test) was performed. Different letters a, b, c, and d indicate that samples are significantly different. The letter “a” marks the lowest value. Higher values are marked with consecutive letters of the alphabet. Bars that share the same letter within the group are not significantly different. Abbreviations as in Figure 1.

Lighting Conditions	Plant/Treatment	Plant Pigments [mg g^−1^ FW]			
		Chlorophyll a	Chlorophyll b	Carotenoids	Anthocyanins
S-CC	C	2.786 ± 0.183 d	0.675 ± 0.104 d	1.127 ± 0.093 d	0.947 ± 0.069 c
	TR	1.826 ± 0.191 bc	0.423 ± 0.059 abc	0.746 ± 0.093 abc	0.658 ± 0.174 ab
	P	2.013 ± 0.084 c	0.539 ± 0.024 bcd	0.861 ± 0.066 abcd	0.799 ± 0.117 bc
	TR+P	2.534 ± 0.049 d	0.582 ± 0.012 cd	1.071 ± 0.018 cd	0.844 ± 0.004 bc
CC174	C	1.591 ± 0.066 abc	0.374 ± 0.082 abc	0.842 ± 0.071 abcd	0.700 ± 0.100 ab
	TR	1.531 ± 0.286 abc	0.262 ± 0.030 a	0.672 ± 0.063 ab	0.615 ± 0.101 ab
	P	1.699 ± 0.109 abc	0.287 ± 0.030 ab	0.799 ± 0.224 abcd	0.743 ± 0.182 abc
	TR+P	1.455 ± 0.315 ab	0.234 ± 0.055 a	0.546 ± 0.106 a	0.549 ± 0.098 a
CC600	C	1.283 ± 0.394 a	0.280 ± 0.081 a	0.676 ± 0.169 ab	0.665 ± 0.093 ab
	TR	1.222 ± 0.267 a	0.368 ± 0.126 abc	0.998 ± 0.170 bcd	0.773 ± 0.122 abc
	P	1.186 ± 0.099 a	0.263 ± 0.032 a	0.657 ± 0.042 ab	0.843 ± 0.111 bc
	TR+P	1.604 ± 0.092 abc	0.348 ± 0.049 abc	0.800 ± 0.008 abcd	0.538 ± 0.208 a

**Table 3 ijms-25-04846-t003:** The content of volatile compounds identified in the vapours of *T. vulgaris*. Abbreviations as in Figure 1; nd—not detected. Data are expressed as the relative peak area (in percentage) of each compound and presented as mean ± SD of three replications, *n* = 3.

Light Conditions/Treatment	Compound Name								Total [% of Volatile Compounds]
	α-Pinene	Myrcene	α-Terpinene	γ-Terpinene	1,8-Cineole	Thymol	Carvacrol	Eugenol	
S-CC/C	13.95 ± 0.34	4.18 ± 0.06	35.50 ± 1.23	5.88 ± 0.21	nd	1.70 ± 0.01	0.62 ± 0.04	2.49 ± 0.09	64.32
S-CC/TR	10.64 ± 0.21	4.32 ± 0.83	21.99 ± 1.61	8.11 ± 0.63	nd	4.08 ± 0.04	nd	2.04 ± 0.18	51.18
S-CC/P	13.44 ± 0.65	3.85 ± 0.22	16.78 ± 0.62	4.06 ± 0.08	nd	2.87 ± 0.14	0.93 ± 0.07	3.85 ± 0.04	45.78
S-CC/P+TR	13.21 ± 0.09	3.97 ± 0.90	18.23 ± 0.71	4.28 ± 0.06	nd	3.04 ± 0.07	nd	2.97 ± 0.09	45.70
CC174/C	13.17 ± 0.59	7.60 ± 1.07	41.71 ± 1.61	12.56 ± 0.44	4.66 ± 0.23	4.66 ± 0.00	nd	0.65 ± 0.03	84.99
CC174/TR	12.98 ± 0.08	6.42 ± 0.31	36.79 ± 1.22	15.28 ± 0.76	nd	4.38 ± 0.02	nd	0.61 ± 0.08	76.46
CC174/P	9.10 ± 0.12	5.40 ± 0.07	33.86 ± 1.58	16.43 ± 0.53	nd	4.13 ± 0.04	nd	0.56 ± 0.03	69.48
CC174/P+TR	11.71 ± 0.53	5.30 ± 0.04	31.44 ± 0.76	12.60 ± 0.48	nd	5.31 ± 0.16	nd	0.65 ± 0.04	67.01
CC600/C	10.87 ± 0.35	6.78 ± 0.27	43.03 ± 0.98	15.38 ± 0.11	nd	2.70 ± 0.11	nd	0.38 ± 0.06	79.14
CC600/TR	10.41 ± 0.99	5.30 ± 0.06	36.39 ± 0.54	14.97 ± 0.86	nd	4.18 ± 0.19	nd	0.34 ± 0.05	71.59
CC600/P	9.87 ± 0.93	4.91 ± 0.11	34.44 ± 0.69	13.36 ± 0.07	nd	5.14 ± 0.47	nd	0.42 ± 0.03	68.13
CC600/P+TR	8.51 ± 0.18	5.47 ± 0.32	35.43 ± 0.43	15.83 ± 0.14	nd	3.91 ± 0.39	nd	0.41 ± 0.00	69.56

**Table 4 ijms-25-04846-t004:** The content of volatile compounds identified in the vapours of *T. serpyllum*. Abbreviations as in Figure 1; nd—not detected. Data are expressed as the relative peak area (in percentage) of each compound and presented as mean ± SD of three replications, *n* = 3.

Light Conditions/Treatment	Compound Name								Total [% of Volatile Compounds]
	α-Pinene	Myrcene	α-Terpinene	γ-Terpinene	1,8-Cineole	Thymol	Carvacrol	Eugenol	
S-CC/C	18.89 ± 0.99	6.16 ± 0.67	23.29 ± 0.64	4.72 ± 0.07	nd	2.60 ± 0.02	0.53 ± 0.34	1.30 ± 0.12	57.49
S-CC/TR	16.23 ± 0.85	5.08 ± 0.07	15.98 ± 1.20	4.78 ± 0.00	nd	3.54 ± 0.07	0.65 ± 0.11	1.35 ± 0.01	47.61
S-CC/P	19.15 ± 0.07	5.80 ± 0.28	24.34 ± 0.75	5.84 ± 0.02	nd	3.31 ± 0.09	nd	1.35 ± 0.05	59.79
S-CC/P+TR	18.20 ± 1.03	5.70 ± 0.01	20.90 ± 0.09	5.18 ± 0.02	nd	3.13 ± 0.02	0.59 ± 0.03	1.76 ± 0.13	55.46
CC174/C	18.81 ± 1.01	7.59 ± 0.81	21.09 ± 1.55	9.46 ± 2.45	1.55 ± 0.73	5.71 ± 0.76	2.91 ± 1.37	1.11 ± 0.09	68.23
CC174/TR	18.91 ± 0.87	6.70 ± 0.04	18.64 ± 0.68	15.44 ± 0.64	nd	5.67 ± 0.09	nd	0.97 ± 0.01	66.33
CC174/P	27.92 ± 1.10	8.98 ± 0.22	16.36 ± 0.12	10.33 ± 1.10	5.31 ± 1.77	6.61 ± 0.51	nd	0.60 ± 0.01	76.11
CC174/P+TR	24.98 ± 0.74	7.90 ± 0.31	27.98 ± 0.34	16.57 ± 0.81	nd	4.29 ± 0.23	0.85 ± 0.40	0.37 ± 0.04	82.94
CC600/C	13.29 ± 0.62	5.32 ± 0.06	32.17 ± 1.65	11.24 ± 0.86	nd	3.99 ± 0.04	nd	0.67 ± 0.08	66.68
CC600/TR	20.49 ± 0.09	6.94 ± 0.21	19.46 ± 0.08	10.00 ± 0.02	nd	5.42 ± 0.08	nd	0.59 ± 0.05	62.90
CC600/P	18.65 ± 0.05	5.29 ± 0.10	12.85 ± 0.07	10.34 ± 1.53	nd	2.56 ± 0.07	nd	0.91 ± 0.03	44.11
CC600/P+TR	23.98 ± 1.01	9.70 ± 0.90	32.34 ± 1.55	13.27 ± 0.72	nd	4.91 ± 0.61	nd	0.25 ± 0.00	84.45

**Table 5 ijms-25-04846-t005:** Changing parameters during the cultivation of thyme plants in a semi-controlled chamber were obtained from the IMGW website (https://meteo.imgw.pl accessed on 10 February 2024).

Cultivation Month	Average Temperature [°C]	Average Air Humidity [%]	Sun Hours	Sun Days
1	20	70	274	17
2	20	70	234	18
3	19	70	165	15

## Data Availability

Data will be made available on request.

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
