# Peer review of "Effect of Light Conditions, Trichoderma Fungi and Food Polymers on Growth and Profile of Biologically Active Compounds in Thymus vulgaris and Thymus serpyllum"

_ijms, 2024, doi:10.3390/ijms25094846_

Round 1
Reviewer 1 Report
Comments and Suggestions for Authors
1.The introduction part needs to be rewritten describing the main aim and objectives. It is unnecessarily too long. Trichoderma part reads like a mini review. Please give necessary reference to relevant work and summarize why you have chosen to see its effect.
2. The plants used belong to any particular cultivar or they are just general species, do mention. The light intensity used had a wide gap 170 and 600. any reason ?
3. In discussion/summary ,can you recommend cultivating the wild thyme relative for cultivation based on your results?
3. What about the cost effectiveness of the used treatments?
4. Summary and conclusion must be improved bringing out the main points.
5. Results and discussion separated will bring out clearly the findings and analysis of the results.
Comments on the Quality of English LanguageOkay.
Author Response
Dear Reviewer,
I would like to express my sincere gratitude to you for taking the time to analyze the article and provide a thorough review. Your feedback has been invaluable in improving the quality of our publication. We have carefully considered all of your remarks and made the necessary modifications to the text, using the track changes mode. Please find our responses to the comments below.
Detailed responses to Reviewer #1:
Reviewer’s comment: The introduction part needs to be rewritten describing the main aim and objectives. It is unnecessarily too long. Trichoderma part reads like a mini review. Please give necessary reference to relevant work and summarize why you have chosen to see its effect.
Response: Thank you for your valuable comments. Following this suggestion, the introduction has been rewritten and shortened. In particular, the sections on the medicinal properties of thyme and the properties of Trichoderma fungi were condensed. Additionally, we have included information on the stimulation of secondary metabolism by stress factors and the potential response to excess light. Furthermore, the objective of the research was elucidated, with the assertion that modifying the growth and development parameters of selected thyme varieties could potentially enhance the content of desired biologically active compounds.
Reviewer’s comment: The plants used belong to any particular cultivar or they are just general species, do mention. The light intensity used had a wide gap 170 and 600. any reason?
Response: The plants belong to the general species, Thymus vulgaris and Thymus serpyllum. We have added this information to the manuscript. We have also added the information about the seeds producer and the names of thyme: Thymus vulgaris ‘Sunshine’ and Thymus serpyllum ‘Pinklilac’.
In terms of lighting, we decided to use two different light intensities, one to support photosynthesis and the other to support secondary metabolism and, in some plants, close to the light intensity that induces stress responses. We added this information in the manuscript.
A photon flux density of 174-175 µmol-m-2 s-1 was considered optimal for the cultivation of other plants:
Bredmose, N. (1993). Effects of year-round supplementary lighting on shoot development, flowering and quality of two glasshouse rose cultivars. Scientia horticulturae, 54(1), 69-85.
Pham, D. M., & Chun, C. (2020). Growth and leaf injury in tomato plants under continuous light at different settings of constant and diurnally varied photosynthetic photon flux densities. Scientia Horticulturae, 269, 109347.
Skłodowska, M., Świercz-Pietrasiak, U., Krasoń, M., Chuderska, A., & Nawrocka, J. (2023). New Insight into Short Time Exogenous Formaldehyde Application Mediated Changes in Chlorophytum comosum L.(Spider Plant) Cellular Metabolism. Cells, 12(2), 232.
A photon flux density of 600 µmol-m-2 s-1 was considered to be the threshold for radiation to have a positive effect on plant growth and metabolism, above which stress responses can be observed in different plants:
Kelly, N., Choe, D., Meng, Q., & Runkle, E. S. (2020). Promotion of lettuce growth under an increasing daily light integral depends on the combination of the photosynthetic photon flux density and photoperiod. Scientia Horticulturae, 272, 109565.
Ashrostaghi, T., Aliniaeifard, S., Shomali, A., Azizinia, S., Abbasi Koohpalekani, J., Moosavi-Nezhad, M., & Gruda, N. S. (2022). Light intensity: The role player in cucumber response to cold stress. Agronomy, 12(1), 201.
Samuolienė, G., Brazaitytė, A., Jankauskienė, J., Viršilė, A., Sirtautas, R., Novičkovas, A., ... & Duchovskis, P. (2013). LED irradiance level affects growth and nutritional quality of Brassica microgreens. Central European Journal of Biology, 8, 1241-1249.
Reviewer’s comment: In discussion/summary ,can you recommend cultivating the wild thyme relative for cultivation based on your results?
Response: Thymus serpyllum is known for its resilience and ability to thrive in a variety of conditions. Based on our results we can recommend the cultivation of wild thyme in a growing chamber with photon flux density of 174 µmol-m-2 s-1. Light of this intensity was sufficient to achieve high biomass and also effectively increased the content of desired secondary metabolites. It is also possible that the addition of food polymers, including alginates and gums and spores of Trichoderma, rather separately, than in combination, may have a positive effect on the synthesis of valuable metabolites of wild thyme.
Reviewer’s comment: What about the cost effectiveness of the used treatments?
Response: The use of Trichoderma spores and food polymers like alginates iand gums in agriculture can be cost-effective when considering their potential benefits in enhancing plant growth, improving soil health, and reducing the need for chemical inputs. Trichoderma are well-known biocontrol agents that can suppress various plant pathogens, including fungi, bacteria, and nematodes. By applying Trichoderma spores to the soil, farmers can reduce the incidence of disease, resulting in higher crop yields and better quality produce. Alginate-based polymers and gums are relatively inexpensive as they are food industry waste and can be used as soil amendments to improve soil structure, water retention and nutrient availability. Unlike synthetic soil conditioners, they are biodegradable and environmentally friendly, contributing to sustainable agricultural practices.
Reviewer’s comment: Summary and conclusion must be improved bringing out the main points.
Response: Thank Yoy for this comment. We have done our best to summarise the results and draw conclusions that may help to improve growing conditions for herbs with health-promoting potential.
Reviewer’s comment: Results and discussion separated will bring out clearly the findings and analysis of the results.
Response: We are grateful for this comment. The results and discussion have been separated. Furthermore, the conclusions of practical application have been highlighted.

Reviewer 2 Report
Comments and Suggestions for Authors
The authors in the work "Effect of Light Conditions, Trichoderma fungi and Food Polymers on Growth and Profile of Biologically Active Compounds in Thymus vulgaris and Thymus serpyllum" tried to study the effect of different lighting conditions and soil treatment, in particular the use of food polymers alone and in combination with spores Trichoderma consortium, on the growth and development of herbs - Thymus vulgaris and Thymus serpyllum. The main goal was to identify optimal conditions for the accumulation of valuable secondary metabolites by plants.
This question is relevant and interesting, and many studies around the world have been aimed at studying it.
However, the presented manuscript raises more questions for me than answers.
About the results obtained by the authors:
We know that the synthesis of secondary antioxidant metabolites and, first of all, phenols, is a stress response of a plant trying to maintain its viability. The plant activates secondary metabolism - which means it doesn’t feel very good! In the context of the manuscript, the authors make no mention of this anywhere other than citing references 44, 45, which state that “anthocyanins have protective functions in plants, including protection from excessive UV radiation and the formation of an internal optical filter that absorbs excess light.” The stress effect of excess light is indicated by a decrease in the content of chlorophylls and an increased content of carotenoids and anthocyanins. But nowhere else is it mentioned that stress can stimulate secondary metabolism!
The pigment complex of the studied species in the experiment practically does not reflect the influence of fungi or polymers. The stress of excess light is already enough for these plants.
Тhe wavelengths of light used by the authors vary greatly. There is no explanation anywhere why the authors used these particular wavelengths. What is their choice based on? They both provide excess light.
Judging by the data obtained, the species Thymus serpyllum is less resistant to stress than Thymus vulgaris. But the authors say nothing about this and make no assumptions about why this might be so.
About the presentation of material:
The introduction is overloaded with information about the medicinal properties of thyme and and properties of Trichoderma. But the introduction makes no mention at all about the stressful effects of excess lighting, although much attention is paid to the influence of lighting in this work. I would like to advise you to completely rewrite the introduction and place the necessary emphasis in it: Why thyme, stimulation of metabolism by stress, reaction to the stress of excess light, Trichoderma, polymers, complex effects, experimental hypothesis and expected result.
A large amount of experimental data on numerous experimental options requires multivariate statistical analysis, with the help of which it would be possible to systematize the results obtained.
Numerous tables of the same type can bore the reader.
I would also recommend separating the “Results” and “Discussion” sections. In "Results" leave only the actual data obtained. And in the “Discussion” give their interpretation. This will systematize the article and make it easier for the reader to understand.
As for eugenol, the reaction of any biologically active substance to stress is not unambiguous. And you are experiencing the effects of a complex of various stressors on plants. Here, for example, is an interesting article by Chinese and German authors, which will help you understand a little about the mechanisms of eugenol accumulation: «Zhao M, Jin J, Wang J, Gao T, Luo Y, Jing T, Hu Y, Pan Y, Lu M, Schwab W, Song C. Eugenol functions as a signal mediating cold and drought tolerance via UGT71A59-mediated glucosylation in tea plants. Plant J 2022 Mar;109(6):1489-1506. doi: 10.1111/tpj.15647». No, I am not a co-author of this article. I'm just pointing out its possible benefits for authors.
Author Response
Dear Reviewer,
I would like to express my sincere gratitude to you for taking the time to analyze the article and provide a thorough review. Your feedback has been invaluable in improving the quality of our publication. We have carefully considered all of your remarks and made the necessary modifications to the text, using the track changes mode. Please find our responses to the comments below.
Detailed responses to Reviewer #2:
Reviewer’s comment: We know that the synthesis of secondary antioxidant metabolites and, first of all, phenols, is a stress response of a plant trying to maintain its viability. The plant activates secondary metabolism - which means it doesn’t feel very good! In the context of the manuscript, the authors make no mention of this anywhere other than citing references 44, 45, which state that “anthocyanins have protective functions in plants, including protection from excessive UV radiation and the formation of an internal optical filter that absorbs excess light.” The stress effect of excess light is indicated by a decrease in the content of chlorophylls and an increased content of carotenoids and anthocyanins. But nowhere else is it mentioned that stress can stimulate secondary metabolism!
Response: Thank you for your valuable comments. According to this recommendation the part of the text has been redrafted. In introduction we have included information on the stimulation of secondary metabolism by stress factors and the potential plant’s response to excess light. We have emphasized that higher-intensity lighting, similar to ultraviolet radiation, is an abiotic stress factor that activates secondary metabolism pathways in plants. This is a well-established fact that has been extensively researched and documented. We agree, that it is important to consider the potential impact of this stress factor on plant growth and development. The flavonoids discussed in this paper, which belong to a broad group of phenolic compounds, may have a particularly protective function against excessive light. Anthocyanins, which are flavonoid glycosides, play a crucial role in protecting the photosynthetic apparatus of plants from damage caused by excessive light.
Reviewer’s comment: The pigment complex of the studied species in the experiment practically does not reflect the influence of fungi or polymers. The stress of excess light is already enough for these plants.
Response: Yes, compared to other parameters, changes in plant pigments content were not as obvious. This may be explained by a number of factors. For instance, the concentration of flavonoids in the tested plants increased under the influence of higher light intensity, which may consequently reduce the risk of photodamage to chlorophyll molecules. The influence of the addition of polymers and spores of Trichoderma fungi was not so obvious. However, in the current version of the manuscript, we attempt to highlight certain changes and detect certain trends, which are consistent with the results of the two-way analysis of variance. The results of this analysis have been added to the Supplementary Material.
Reviewer’s comment: Тhe wavelengths of light used by the authors vary greatly. There is no explanation anywhere why the authors used these particular wavelengths. What is their choice based on? They both provide excess light.
Response: In terms of lighting, we decided to use two different light intensities, one to support photosynthesis and the other to support secondary metabolism and, in some plants, close to the light intensity that induces stress responses. We added this information in the manuscript. Lines:
A photon flux density of 174-175 µmol-m-2 s-1 was considered optimal for the cultivation of other plants:
Bredmose, N. (1993). Effects of year-round supplementary lighting on shoot development, flowering and quality of two glasshouse rose cultivars. Scientia horticulturae, 54(1), 69-85.
Pham, D. M., & Chun, C. (2020). Growth and leaf injury in tomato plants under continuous light at different settings of constant and diurnally varied photosynthetic photon flux densities. Scientia Horticulturae, 269, 109347.
Skłodowska, M., Świercz-Pietrasiak, U., Krasoń, M., Chuderska, A., & Nawrocka, J. (2023). New Insight into Short Time Exogenous Formaldehyde Application Mediated Changes in Chlorophytum comosum L. (Spider Plant) Cellular Metabolism. Cells, 12(2), 232.
A photon flux density of 600 µmol-m-2 s-1 was considered to be the threshold for radiation to have a positive effect on plant growth and metabolism, above which stress responses can be observed in different plants:
Kelly, N., Choe, D., Meng, Q., & Runkle, E. S. (2020). Promotion of lettuce growth under an increasing daily light integral depends on the combination of the photosynthetic photon flux density and photoperiod. Scientia Horticulturae, 272, 109565.
Ashrostaghi, T., Aliniaeifard, S., Shomali, A., Azizinia, S., Abbasi Koohpalekani, J., Moosavi-Nezhad, M., & Gruda, N. S. (2022). Light intensity: The role player in cucumber response to cold stress. Agronomy, 12(1), 201.
Samuolienė, G., Brazaitytė, A., Jankauskienė, J., Viršilė, A., Sirtautas, R., Novičkovas, A., ... & Duchovskis, P. (2013). LED irradiance level affects growth and nutritional quality of Brassica microgreens. Central European Journal of Biology, 8, 1241-1249.
Reviewer’s comment: Judging by the data obtained, the species Thymus serpyllum is less resistant to stress than Thymus vulgaris. But the authors say nothing about this and make no assumptions about why this might be so.
Both Thymus serpyllum and Thymus vulgaris demonstrate a certain degree of resilience to abiotic stress factors. However, their responses to these factors may differ. Thymus serpyllum is generally better adapted to harsh conditions, as it is able to grow in more challenging environments, such as on rocks. It may also exhibit enhanced secondary metabolite production under excessive light stress compared to Thymus vulgaris. The results of this study indicate that it is challenging to definitively determine which species of thyme exhibited greater resistance to stress associated with intense lighting. However, it can be recommended that light with an intensity of 174 µmol-m-2 s-1 be used to stimulate the growth and synthesis of the desired secondary metabolites for both species.
Reviewer’s comment: The introduction is overloaded with information about the medicinal properties of thyme and properties of Trichoderma. But the introduction makes no mention at all about the stressful effects of excess lighting, although much attention is paid to the influence of lighting in this work. I would like to advise you to completely rewrite the introduction and place the necessary emphasis in it: Why thyme, stimulation of metabolism by stress, reaction to the stress of excess light, Trichoderma, polymers, complex effects, experimental hypothesis and expected result.
Response: Thank you for a very helpful suggestion. Following this suggestion, the introduction has been rewritten and shortened. In particular, the sections on the medicinal properties of thyme and the properties of Trichoderma fungi were condensed. Additionally, we have included information on the stimulation of secondary metabolism by stress factors and the potential response to excess light. Furthermore, the objective of the research was elucidated, with the assertion that modifying the growth and development parameters of selected thyme varieties could potentially enhance the content of desired biologically active compounds.
Reviewer’s comment: A large amount of experimental data on numerous experimental options requires multivariate statistical analysis, with the help of which it would be possible to systematize the results obtained.
Response: Thank you for this comment. Our previous description did not indicate that such an analysis was performed. In the current version of the manuscript, we have added information that a two-way ANOVA analysis was performed. We have described the results according to the statistical results included in the graphs and added in the Supplementary Materials.
Reviewer’s comment: Numerous tables of the same type can bore the reader.
I would also recommend separating the “Results” and “Discussion” sections. In "Results" leave only the actual data obtained. And in the “Discussion” give their interpretation. This will systematize the article and make it easier for the reader to understand.
Response: We are grateful for this comment. The results and discussion have been separated. Furthermore, the conclusions of practical application have been highlighted.
Reviewer’s comment: As for eugenol, the reaction of any biologically active substance to stress is not unambiguous. And you are experiencing the effects of a complex of various stressors on plants. Here, for example, is an interesting article by Chinese and German authors, which will help you understand a little about the mechanisms of eugenol accumulation: «Zhao M, Jin J, Wang J, Gao T, Luo Y, Jing T, Hu Y, Pan Y, Lu M, Schwab W, Song C. Eugenol functions as a signal mediating cold and drought tolerance via UGT71A59-mediated glucosylation in tea plants. Plant J 2022 Mar;109(6):1489-1506. doi: 10.1111/tpj.15647». No, I am not a co-author of this article. I'm just pointing out its possible benefits for authors.
Response: Thank you for this valuable comment. We read the article you recommended and also found another valuable publication on the role of eugenol in plant defence mechanisms in response to abiotic stress. Both articles allowed us to enrich our manuscript and shed light on the importance of eugenol in plant immunity.
We have enriched our manuscript with the two references mentioned:
Zhao M, Jin J, Wang J, Gao T, Luo Y, Jing T, Hu Y, Pan Y, Lu M, Schwab W, Song C. Eugenol functions as a signal mediating cold and drought tolerance via UGT71A59-mediated glucosylation in tea plants. Plant J. 2022 Mar;109(6):1489-1506.
Xu J, Wang T, Sun C, Liu P, Chen J, Hou X, Yu T, Gao Y, Liu Z, Yang L, Zhang L. Eugenol improves salt tolerance via enhancing antioxidant capacity and regulating ionic balance in tobacco seedlings. Front Plant Sci. 2024 Jan 16;14:1284480.

Reviewer 3 Report
Comments and Suggestions for Authors
The research work is useful and interesting.
I have the following minor comments to the authors:
1. Is the biomass measurement taken in terms of dry weight or fresh weight?
2. The contents of all pigments did not show any pattern with the Trichoderma fungi / food polymers or both under different light conditions. This is bit strange.
3. Among T. vulgaris and T. serpyllume, which one showed overall better performance in all conditions? Please mention that in the abstract and conclusion.
4. There are minor typing errors and grammatical mistakes. Please check it.
Comments on the Quality of English LanguageMinor editing is required.
Author Response
Dear Reviewer,
I would like to express my sincere gratitude to you for taking the time to analyze the article and provide a thorough review. Your feedback has been invaluable in improving the quality of our publication. We have carefully considered all of your remarks and made the necessary modifications to the text, using the track changes mode. Please find our responses to the comments below.
Detailed responses to Reviewer #3:
Reviewer’s comment: Is the biomass measurement taken in terms of dry weight or fresh weight?
Response: In the course of the experiments, the fresh weight of the plants was weighed immediately after harvest.
Reviewer’s comment: The contents of all pigments did not show any pattern with the Trichoderma fungi / food polymers or both under different light conditions. This is bit strange.
Response: Yes, compared to other parameters, changes in plant pigments content were not as obvious. This may be explained by a number of factors. For example because the concentration of flavonoids in the tested plants increases under the influence of higher intensity lighting, it may reduce the risk of photodamage to chlorophyll molecules. However, in the current version of the manuscript, we try to highlight certain changes and detect certain trends, consistent with the results of the two-way analysis of variance. We have included the results of this analysis in the Supplementary Material.
Reviewer’s comment: Among T. vulgaris and T. serpyllume, which one showed overall better performance in all conditions? Please mention that in the abstract and conclusion.
Response: Thank you for this suggestion. Both Thymus serpyllum and Thymus vulgaris demonstrate a certain degree of resilience to abiotic stress factors. However, their responses to these factors may differ. Thymus serpyllum is generally better adapted to harsh conditions, as it is able to grow in more challenging environments, such as on rocks. It may also exhibit enhanced secondary metabolite production under excessive light stress compared to Thymus vulgaris. The results of this study indicate that it is challenging to definitively determine which species of thyme exhibited greater resistance to stress associated with intense lighting. However, it can be recommended that light with an intensity of 174 µmol-m-2 s-1 be used to stimulate the growth and synthesis of the desired secondary metabolites for both species.
Reviewer’s comment: There are minor typing errors and grammatical mistakes. Please check it.
Response: Thank you for this comment. We have proofread the manuscript for grammatical, stylistic and punctuation errors.

Round 2
Reviewer 2 Report
Comments and Suggestions for Authors
The authors have significantly improved the presentation of their results. I still get bored with the same type of graphs, but the material is now presented much more clearly than before. The authors also explained many methodological points that were previously not presented well enough or were missed altogether.